# Glucose Starvation or Pyruvate Dehydrogenase Activation Induce a Broad, ERK5-Mediated, Metabolic Remodeling Leading to Fatty Acid Oxidation

**DOI:** 10.3390/cells11091392

**Published:** 2022-04-20

**Authors:** Abrar Ul Haq Khan, Hamideh Salehi, Catherine Alexia, Jose M. Valdivielso, Milica Bozic, Isabel C. Lopez-Mejia, Lluis Fajas, Sabine Gerbal-Chaloin, Martine Daujat-Chavanieu, Delphine Gitenay, Martin Villalba

**Affiliations:** 1IRMB, University of Montpellier, INSERM, 34295 Montpellier, France; catherine.alexia@inserm.fr (C.A.); sabine.gerbal-chaloin@inserm.fr (S.G.-C.); martine.daujat@inserm.fr (M.D.-C.); delphine.gitenay@umontpellier.fr (D.G.); 2LBN, University of Montpellier, 34295 Montpellier, France; s_hamideh@yahoo.com; 3Vascular and Renal Translational Research Group, Biomedicine Research Institute of Lleida (IRBLLIDA), 25198 Lleida, Spain; josemanuel.valdivielso@udl.cat (J.M.V.); bozicm@medicina.udl.cat (M.B.); 4Center for Integrative Genomics, University of Lausanne, CH-1015 Lausanne, Switzerland; isabel.lopezmejia@unil.ch (I.C.L.-M.); lluis.fajas@unil.ch (L.F.); 5IGMM, CNRS, INSERM, 34295 Montpellier, France; 6IRMB, University of Montpellier, INSERM, CNRS, CHU Montpellier, 34295 Montpellier, France; 7Institut du Cancer Avignon-Provence Sainte Catherine, F-84000 Avignon, France

**Keywords:** fatty acid oxidation, glycolysis, ERK5, metabolic flexibility, metabolic plasticity

## Abstract

Cells have metabolic flexibility that allows them to adapt to changes in substrate availability. Two highly relevant metabolites are glucose and fatty acids (FA), and hence, glycolysis and fatty acid oxidation (FAO) are key metabolic pathways leading to energy production. Both pathways affect each other, and in the absence of one substrate, metabolic flexibility allows cells to maintain sufficient energy production. Here, we show that glucose starvation or sustained pyruvate dehydrogenase (PDH) activation by dichloroacetate (DCA) induce large genetic remodeling to propel FAO. The extracellular signal-regulated kinase 5 (ERK5) is a key effector of this multistep metabolic remodeling. First, there is an increase in the lipid transport by expression of low-density lipoprotein receptor-related proteins (LRP), e.g., CD36, LRP1 and others. Second, an increase in the expression of members of the acyl-CoA synthetase long-chain (ACSL) family activates FA. Finally, the expression of the enzymes that catalyze the initial step in each cycle of FAO, i.e., the acyl-CoA dehydrogenases (ACADs), is induced. All of these pathways lead to enhanced cellular FAO. In summary, we show here that different families of enzymes, which are essential to perform FAO, are regulated by the signaling pathway, i.e., MEK5/ERK5, which transduces changes from the environment to genetic adaptations.

## 1. Introduction

Cells adapt to a changing environment by adjusting their metabolism [1]. An initial plastic adaptation should be compensated or reinforced by genetic means to allow long-term fitness in the new environment [1]. A prototype of adaption is the metabolic rewiring of mammalian cells during changes in substrate and/or oxygen availability. This is known as metabolic flexibility and complements metabolic plasticity, which resumes the series of metabolic phenotypes present in a cell. A situation where metabolic flexibility and plasticity have largely been investigated is the Warburg effect, a process by which tumor cells produce lactate from glucose instead of fully oxidizing it, even in the presence of ample amounts of oxygen [2].

Two of the main metabolic pathways leading to energy production are glycolysis and fatty acid oxidation (FAO) [3]. Oxidation of either glucose or fatty acids (FA) may preponderate depending on nutritional and/or physiologic conditions. Logically, both pathways affect each other, with lipolysis inhibiting glycolysis and glycolysis decreasing FAO [3]. It is expected that inhibition of one pathway will stimulate the other, and interestingly, Hsieh and colleagues have shown in yeast that glucose starvation induces fat metabolism genes [4].

Dichloroacetate (DCA), a pyruvate dehydrogenase kinase 1 (PDK1) inhibitor, inhibits glycolysis and favors oxidative phosphorylation (OXPHOS) [5]. DCA was used to control lactic acidosis [6] and some trials also suggested its value in hypercholesterolemia treatment [7]. DCA controls cholesterol homeostasis [8] and lipid hepatic metabolism in septic mice [9]. Indeed, it decreases plasma cholesterol and triglyceride (TG) levels in animal models and humans [10]. This effect is consecutive to an increase in TG oxidation [11]. Clinical trials with DCA showed interesting results in cholesterol levels; however, its use was discontinued due to a reversible neuropathological toxicity [12,13]. By promoting OXPHOS, DCA activates the extracellular signal-regulated kinase 5 (ERK5) pathway that turns on the transcription factor family MEF2 [14]. MEF2 binds to the LDLR promoter, inducing de novo LDLR expression, which is responsible for sequestering cholesterol into cells [8]. Other physiological effects of DCA are mediated by ERK5, also mainly through MEF2 activation [15,16]. Although ERK5 also has its own transcriptional domain, it translates stress signals, including metabolic stress, to gene expression through MEF2 [17,18,19,20,21]. ERK5 plays key roles multiple cellular functions [20], including on cell metabolism [22] and to some extent in the control of lipid metabolism [8,23,24]. This could explain the role of ERK5 in atherosclerosis [19,23,25,26], a disease in which cholesterol levels constitute a risk factor. In this study, we set to determine if ERK5 could be a good candidate to modulate metabolic flexibility by sensing glucose starvation and preparing cells to switch to FAO.

## 2. Materials and Methods

### 2.1. Ethical Statement

Experimental procedures were conducted according to the European guidelines for animal welfare (2010/63/EU). Protocols were approved by the Animal Care and Use Committee “Languedoc-Roussillon” (approval number: CEEA-LR-12163). All methods were carried out in accordance with the approved guidelines and regulations of this committee.

### 2.2. In Vivo Mouse Experiments

B6 wt mice were treated with a daily single dose of DCA (50 mg/kg/day) intraperitoneally, and mouse mRNA was analyzed in spleen and liver at different time points.

Experiments involving engraftment of AML cells were carried out using 6- to 8-week-old male NSG mice as previously described [8,14,16]. For engraftment of human cells, 1 million AML cells were injected intravenously (i.v.) through the lateral tail vein in non-irradiated mice. NSG mice with established human AML tumors (day 80 post-graft) were treated with DCA (50 mg/kg, 1 dose/day by gavage, starting at day 1 for 16 consecutive days). Human tumor AML cells gather in mouse spleen and bone marrow; hence, we isolated mRNA from these organs. We used human-specific primers to visualize expression of human mRNA.

For the high-fat-diet experiments, C57BL/6J mice (12 weeks old) were bred and housed in pathogen-free conditions in the animal facility of the University of Lleida (Lleida, Spain). Animals were kept in a 12 h light/dark cycle at 22 °C with ad libitum access to food and water. At 12 weeks of age, mice were placed on a high-fat diet (HFD) or “Paigen diet” (1.25% Cholesterol, 0.5% Cholic acid, 15% Cocoa butter, 1% Corn oil, S9358-E030, Ssniff, Soest, Germany) for 4 weeks. Two weeks after the beginning of HFD feeding, mice were subjected to daily injections of sodium dichloroacetate (Santa Cruz, sc-203275, Dallas, TX, USA) (50 mg/kg/day), intraperitoneally, during the following 2 weeks of the experiment. Mouse experiments performed in this study were approved by the Ethics Committee of the University of Lleida in accordance with the guidelines of European Research Council for the care and use of laboratory animals. Body weight was measured every week from the beginning of the experiment. Individual food intake was measured after 4 weeks of the experiment. At the end of the experiment, blood was collected by cardiac puncture after a 16 h overnight fast. The animals were perfused with PBS through a puncture in the left ventricle and organs were harvested.

### 2.3. Cell Lines and Culture Conditions

The leukemic human cell lines T Jurkat Tag, which carries the SV40 large T Ag, and OCI-AML3 were grown in RPMI 1640–Glutamax (GIBCO; Thermofisher Scientific, Waltham, MA, USA) supplemented with 5% or 10% (OCI-AML3) FBS (Jurkat). The primary cell line (BCL-P2) that we derived from a B-cell lymphoma patient [27] was grown in the same medium with 10% FBS. In certain experiments, cells were grown in RPMI 1640 without glucose (GIBCO 11879) with the addition of 2 mM glutamine and 10 mM galactose (OXPHOS medium). HepG2, HepG2-C3A and HuH7 cells were grown in MEM and DMEM supplemented with FBS, sodium pyruvate, glutamine, penicillin and streptomycin, respectively. Cellular confluence during experiments was between 80–85%.

### 2.4. Glucose Starvation

Cells were washed once in cells RPMI 1640 without glucose (GIBCO 11879) and were grown in the same medium with the addition of 2 mM glutamine and 10 mM galactose (OXPHOS medium).

### 2.5. Human Liver Samples and Preparation of PHHs Cultures

Liver samples were obtained from liver resections performed in adult patients for medical reasons. Human hepatocytes isolation and culture were performed as described previously [28]. Briefly, after liver perfusion, hepatocytes were counted, and cell viability was assessed by trypan blue exclusion test. A suspension of 1 × 10^6^ cells/mL per well was added in 12-well plates precoated with type I collagen (Beckton Dickinson, Franklin Lakes, NJ, USA) and cells were allowed to attach for 12 h. Then, the supernatant containing dead cells and debris was carefully removed and replaced with 1 mL of serum-free long-term culture medium (Lanford medium, LNF). The number of confluent attached cells was estimated at ~1.5 × 10^5^ cells/cm^2^.

### 2.6. Reagents and Antibodies

DCA was from Santa Cruz Technologies. Galactose and glutamine were from GIBCO. Human anti-CD36-PE, anti-CD36L-PE, anti-LPR1-PE and IgG were from Beckton Dickinson and 7AAD from Beckman (Brea, CA, USA). The MEK5 inhibitor BIX02189 and the ERK5 inhibitor XMD8–92 were from Selleck Chemicals (Radnor, PA, USA). RIPA buffer to prepare protein extracts was from Euromedex (Souffelweyersheim, France). The complete protease inhibitor cocktail (Complete EDTA-free) and the phosphatase inhibitor cocktail (PhosSTOP) were from Roche (Bale, Switzerland). ERK5, ACDVL and MEF2A/C antibodies were from Cell Signaling Technology (Danvers, MA, USA) and Abcam (Cambridge, UK), respectively. The antibody against β-Actin and HRP-labeled secondary antibodies were from Sigma (Burlington, MA, USA).

### 2.7. Transient Transfection

Jurkat cells in logarithmic growth phase were transfected with the indicated amounts of plasmid by electroporation. In each experiment, cells were transfected with the same total amount of DNA by supplementing with an empty vector. Cells were incubated for 10 min at RT with the DNA mix and electroporated using the Gene Pulser Xcell™ Electroporation system (Bio-Rad; Hercules, CA, USA) at 260 mV, 960 mF in 400 µL of RPMI 1640. Expression of the different proteins was confirmed by Western blot. The transfection efficiency in Jurkat TAg cells is between 60 and 80%. OC-AML-3 cells were transfected using Amaxa TM D-Nucleofector TM Lonza Kit according to the manufacturer’s protocol. In HuH7 and HCT116 cells, transfection of 30–50 nM siRNAs was carried out using Lipofectamine RNAiMAX (Invitrogen, Waltham, MA, USA) in Opti-MEM (Invitrogen) according to the manufacturer’s instructions. Primary hepatocytes were transfected twice, at days 1 and 3 post-seeding. Cells were harvested 48 to 96 h post-transfection.

### 2.8. Plasmids

The expression vectors for ERK5, the pSUPER expression vector for GFP alone or GFP plus shERK5 and the pSiren-retroQ-puro (BD Biosciences) retroviral vectors for shERK5 and control have been previously described [29]. Control, MEF2A and C and ERK5 siRNA were ON-TARGETplus SMARTpools (mixture of 4 siRNA) were from Dharmacon (Lafayette, CO, USA).

### 2.9. Counting and Determination of Cell Viability

Cell number, viability and cell death was analyzed with the Muse Cell Analyzer (Millipore; Burlington, MA, USA) by incubating cells with Muse Count and Viability and Annexin V and Dead Cell kits, respectively, following the manufacturer’s instructions.

### 2.10. RT-QPCR

Total RNA was extracted using NucleoSpin RNA isolation columns (Macherey-Nagel; Düren, Germany), reverse transcription was carried out using iScript™ cDNA Synthesis Kit (Bio-Rad). Quantitative PCR was performed with KAPA SYBR Green qPCR SuperMix (Cliniscience; Nanterre, France)) and a CFX Connect™ Real-Time qPCR machine (BioRad). All samples were normalized to β-actin mRNA levels. Results are expressed relative to control values arbitrarily set at 100.

### 2.11. Immunoblotting

Protein analysis by immunoblotting was performed essentially. Briefly, samples were collected, washed out with PBS and lysed with RIPA buffer. Protein concentration was determined by BCA assay (Pierce; Dallas, TX, USA) before electrophoresis in 4–15% TGX gels (BioRad) and equal amount of protein was loaded in each well. Protein transfer was performed in TransTurbo system (BioRad) in PVDF membranes. After blocking for 1 h with 5% non-fat milk, membranes were incubated overnight at 4 °C in agitation with primary antibodies, washed three times with PBS-Tween 0.1% and incubated with the appropriate HRP-labeled secondary antibody for 1 h. Membranes were washed out three times with PBS-Tween 0.1% and developed with Substrat HRP Immobilon Western (Millipore). Band quantification was performed using the “ImageLab” (version 3.0.1) software from Bio-Rad and represented as the ratio between the protein of interest and a control protein, i.e., actin. The value of 1 is arbitrarily given to control cells. One blot representative of several experiments is shown.

### 2.12. Cellular Lipid Uptake

Following treatment with DCA or OXPHOS medium, cells were incubated with BODIPYTM FL-C12 (Dodecanoic Acid) or BODIPYTM FL-C16 (Hexadecanoic Acid) from InvitrogenTM (Thermofisher) in PBS with 2% FBS and incubated at 37 °C for 30 min. Cells were then washed and suspended in 200–250 μL PBS 2% FBS and fluorescence was analyzed using Gallios flow cytometer (Beckman) and the Kaluza (2.1) software, Beckman.

### 2.13. Serum Triglycerides

They were determined by standard clinical methods using a multichannel Hitachi Modular analyzer (Roche Diagnostics, Indianapolis, IN, USA).

### 2.14. Extracellular Flux Analysis

Mitochondrial respiration and glycolysis was evaluated by using the XF24 or XF96 Flux analyzers (SeaHorse Bioscience; North Billerica, MA, USA), which measures oxygen consumption rate (OCR) and extracellular acidification rate (ECAR). HCT116 were treated with 5 mM DCA over 72 h (48 h in a 10 cm plate, and the last 24 h in the seahorse plate). To account for differences in cell proliferation, 8000 cells were seeded per well in the DCA treatment condition, and 7500 cells were seeded in the control conditions the day before OCR assay in the Seahorse XF96 OCR measurement was performed in XF media (non-buffered DMEM) supplemented with 2.5 mM glucose, 2 mM L-glutamine and 1 mM sodium pyruvate, under basal conditions and in response to mitochondrial inhibitors: 1 µM oligomycin, 0.33 µM FCCP, 100 nM rotenone and 1 µM antimycin A (Sigma). OCR, in pMoles/min, was measured during 4 min. The basal respiration rate was calculated as the difference between basal OCR and OCR after inhibition of mitochondrial complex 1 and 3 with rotenone and antimycin A, respectively. The maximum respiration rate was measured following addition of the uncoupler FCCP (uncoupled rate), indicative of the maximum activity of electron transport and substrate oxidation achievable by the cells. To assess fatty acid oxidation involvement in OCR, cells were treated with 5 µM etomoxir 1 h prior to flux analyses.

For the glycolysis assay, HepG2 cells were treated with 5 mM DCA over 72 h (48 h in a 10 cm plate, and the last 24 h in the seahorse plate). To account for differences in cell proliferation, 60,000 cells were seeded per well in the DCA treatment condition and 50,000 were seeded in the control conditions the day before glycolysis assay in the Seahorse XF24. The day of the assay, the medium was replaced by base DMEM 2 mM Glutamine (no FBS, no Glucose, no Pyruvate). After 3 measurements, 20 mM Glucose was injected. After 6 measurements 100 mM 2DG was injected to completely abrogate glycolysis. The glycolytic rate, or glycolytic capacity, was calculated as the difference between maximal ECAR to basal ECAR. ECAR measurements were normalized with protein concentration (measured by BCA in each well). Statistical analyses were performed with Prism 8 software, GraphPad Software, Inc. (San Diego, CA).

### 2.15. Flow Cytometry

Briefly, 1 × 10^6^ cells were stained with antibody in PBS with 2% FBS and incubated at 37 °C for 30 min. Cells were then washed and suspended in 200–250 μL PBS 2% FBS and staining was analyzed using a Gallios flow cytometer (Beckman) and the Kaluza software.

### 2.16. Statistical Analysis

The statistical analysis of the difference between means of paired samples was performed using the paired *t* test. Analysis of multiple comparisons with single control was performed using one-way ANOVA with a post-hoc Tukey test. The results are given as the confidence interval (* *p* < 0.05, ** *p* < 0.01, *** *p* < 0.005). All the experiments described in the figures with a quantitative analysis were performed at least three times in duplicate. Other experiments were performed three times with similar results.

## 3. Results

### 3.1. Changes in Metabolism Regulate Expression of Proteins Involved in Lipid Uptake through the ERK5 Pathway

A successful metabolic switch firstly implies the intake of the new source of energy, e.g., lipids. Secondly, it implies transport to the site where energy must be produced, e.g., the mitochondria. Finally, the organelle must adapt to the new energy production, e.g., FAO. We wished to investigate how inducing a metabolic switch affects these three parameters: cellular uptake, intake in the organelle and metabolic remodeling. We deprived the OCI-AML3 cells of glucose, and instead supplemented the media with galactose, which allows cells to synthesize nucleic acids through the pentose phosphate pathway, and with glutamine, an amino acid that supports oxidative phosphorylation (OXPHOS) [30,31]. The absence of glucose inhibits glycolysis and forces cells to find an alternative source of energy. In our conditions, OCI-AML3 reacted by increasing the mRNA expression of the FA transporter CD36, as well as its localization at the plasma membrane (Figure 1A). We found a similar increase in other tumor cell lines, i.e., HuH7, HepG2C and Jurkat, as well as in a primary cell line that we derived from a B-cell lymphoma patient (BCL-P2; Figure 1A).

Next, we forced cells to change their metabolism by using the pyruvate dehydrogenase kinase-1 (PDK1) inhibitor dichloroacetate (DCA), which inhibits glycolysis and favors OXPHOS [8,14,16,32]. DCA induced CD36 mRNA and plasma membrane CD36 protein in a large range of tumor cell lines (Figure 1A and Appendix A). DCA also induced CD36 expression in non-transformed cells such as primary hepatocytes (Figure 1A) and primary natural killer (NK) cells (Appendix A). Notably, prolonged DCA treatment to HepG2C cells for one week further augmented the expression of CD36 on the cell surface (Appendix A).

Next, we engrafted human acute myeloid cells (AML) into NSG mice, which were then treated with DCA as previously described [32]. We isolated the bone marrow (BM) and spleen of mice, where AML cells niche, and analyzed CD36 mRNA expression by using qPCR primers specific for human mRNA. DCA increased CD36 mRNA expression of human tumor cells in vivo (Figure 1B).

We had previously shown that DCA-induced metabolic rewiring requires at least partial activation of the ERK5/MEF2 pathway [8,14,16]. We decreased ERK5 or MEF2 proteins by using small interference RNAs (siRNAs). This effectively reduces ERK5 mRNA and protein between 30% and 70% in the different cell types used in this study [8,14,16], which we confirmed below in this study (see Figure 4E). As previously described, ERK5 has multiple functions, e.g., adaptor, transcriptional activator, kinase and differences on ERK5 levels can differently modulate its effects on the expression of its targets [17,18,19,20,21]. These decreases in expression were similar for MEF2 proteins, although the decrease was bigger for MEF2A than for MEF2C [8,14,16]. siRNA for ERK5 or MEF2 reduced CD36 mRNA expression in resting primary (Figure 1C) and in transformed (Appendix A) hepatocytes, and in resting or DCA-treated OCI-AML-3 cells (Figure 1D). ERK5 overexpression in Jurkat cells increased CD36 at the plasma membrane (Appendix A). Additionally, chemical inhibition of ERK5 or MEK5, the upstream MAPKK in the ERK5 pathway, decreased CD36 levels in the plasma membrane in several tumor cells (Appendix A).

Next, we investigated the effect of metabolic rewiring in the expression of other receptors involved in FA intake. The expression of the scavenger receptor class B type 1 (SCARB1 or CD36L1), which functions as a receptor for high-density lipoprotein (HDL) [33], increased in multiple DCA-treated cells and in cells growing in glucose-free medium (Appendix A). Conversely, ERK5 or MEK5 inhibitors decreased its expression (Appendix A).

CD36 and CD36L1 belong to the low-density lipoprotein receptor-related protein (LRP) family, which also includes LRP1, LRP1B, LRP5, LRP6 and LRP10 [34]. Hence, we analyzed if other LRP family members were also regulated by metabolic rewiring. LRP1, which is involved in receptor-mediated endocytosis and lipoprotein metabolism, increased in multiple cell types after metabolic changes similar to those that regulate CD36 and CD36L1 (Figure 1E and Appendix A), including in vivo, in engrafted AML cells (Appendix A). Conversely, decreasing ERK5 or MEF2 expression, or using ERK5 and MEK5 inhibitors, reduced LRP1 expression (Appendix A). The mRNA of other members of the LRP family, such as LRP1B, LRP5, LRP6 and LRP10, were equally upregulated by DCA and downregulated by siERK5 (Appendix A). Interestingly, Lipase C (LIPC), hepatic type, which is involved in extracellular lipid metabolism, was equally regulated by DCA treatment or ERK5 levels (Appendix A). This enzyme converts intermediate-density lipoprotein (IDL) to low-density lipoprotein (LDL), and thus plays an important role in the regulation of blood TG levels [35]. Taken together, these results demonstrate that glucose deprivation in cells upregulates the expression of proteins involved in FA uptake at the transcriptional level.

### 3.2. Changes in Metabolism Regulate Lipid Uptake through the ERK5 Pathway

Next, we investigated the possible physiological relevance of the increased expression of these proteins involved in FA metabolism. DCA or glucose deprivation increased cellular uptake of FAs, i.e., dodecanoic and hexadecenoic acids, in HepG2C hepatocytes (Figure 2A) and in four other cell lines (Appendix A). In accordance with mRNA expression, inhibition of ERK5 or MEK5 decreased FA uptake in BCL-P2 cells (Figure 2B), OCI-AML3 and HepG2 cells (Appendix A).

We further investigated this phenomenon by using a Raman microscope in HuH7 hepatocytes to confirm if metabolic rewiring leads to intracellular lipid accumulation. As expected, we found that DCA treatment in these cells doubled the amount of intracellular lipids (Figure 2C). Moreover, mice fed for 4 weeks with a high-fat diet and treated daily with DCA (50 mg/kg) showed decreased blood triglycerides (TG) compared to non-treated mice (Figure 2D), without affecting body weight or food intake (Appendix A). This further supports the notion that DCA also favors TG intake by cells in vivo.

In summary, these findings suggest that the intake of lipids is largely regulated by metabolic changes through the ERK5 pathway.

### 3.3. Changes in Metabolism Regulate the Expression of Enzymes Involved in Lipid Esterification through the ERK5 Pathway

Next, we analyzed proteins that are involved in FA activation for their use in ATP generation through FAO. The acyl-CoA synthetase long-chain (ACSL) family members catalyze the esterification and activation of the most profuse FA into fatty acyl-CoA, and thereby play a key role in their degradation and indirectly promote their uptake [36]. DCA or glucose-free medium induced ACSL1 mRNA expression in several cell lines and primary transformed and non-transformed cells (Figure 3A and Appendix A). DCA also increased ACSL1 expression in vivo in AML tumor cells engrafted in NSG mice (Figure 3B). Decreasing ERK5 or MEF2 mRNA levels with siRNA reduced ACSL1 mRNA in primary hepatocytes (Figure 3C) and several cell lines (Figure 3D). This was not specific to ACSL1 because ACSL6 mRNA was equally affected by all these treatments in several cell lines and in primary hepatocytes (Figure 3E and Appendix A). These observations suggested that genes involved in lipid esterification were also modulated in an ERK5-dependent fashion in conditions in which glycolysis was suppressed.

### 3.4. Changes in Metabolism Regulate FAO through an ERK5-Dependent Pathway

The acyl-CoA dehydrogenases (ACADs) are enzymes that catalyze the initial step in each cycle of FAO [37]. They are divided into four groups based on their specificity for short (short-chain acyl-CoA dehydrogenase (SCAD), medium (medium-chain acyl-CoA dehydrogenase (ACADM), long-chain (long chain acyl-CoA dehydrogenase (ACADLC) and very long-chain (very-long-chain acyl-CoA dehydrogenase (ACADVL) fatty acid acyl-CoA substrates. Hence, ACADVL, which localizes in the inner mitochondrial membrane, plays an essential role in FAO [38]. DCA and glucose-free medium increased expression of ACADVL1 mRNA and protein in several cell types, including primary cells (Figure 4A,B and Appendix A). DCA also increased ACADVL1 mRNA expression in vivo in AML cells engrafted in NSG mice (Figure 4C). Moreover, wild-type mice treated for up to 3 days with DCA showed increased Acadvl1 mRNA levels in their liver and spleen (Figure 4D). Decreasing ERK5 or MEF2 with siRNA reduced ACADVL1 in several cell types (Figure 4E and Appendix A). Acyl-Coenzyme A dehydrogenase, C-4 to C-12 straight chain (ACADM), which targets medium-chain FA, was equally regulated by DCA, by the absence of glucose or by ERK5 levels (Appendix A).

As previously discussed, DCA inhibits glycolysis and the results presented above suggest that it induces FAO. In agreement, it increased the basal oxygen consumption rate (OCR; Figure 5A) and decreased glucose-induced extracellular acidification rate (ECAR; Figure 5B), likely because it inhibits aerobic glycolysis and this decreases lactate production. In contrast, DCA did not increase maximal respiration (Figure 5A), suggesting that DCA did not increase mitochondrial mass. Blocking FAO with etomoxir largely decreased basal respiration, showing that HCT116 cells largely depended on FAO (Figure 5A). DCA could still increase OCR in the presence of etomoxir; hence, DCA also increased oxidative phosphorylation (OXPHOS) of other substrates as previously shown [8,14,16,30]. Finally, etomoxir decreased expression of the FA transporter CD36, CD36L and LRP1 (Figure 5C), suggesting that inhibition of FAO leads to decreased FA uptake.

## 4. Discussion

Cells should adapt to their environment, which includes substrate availability. This is obvious for unicellular organisms, but it also applies to cells in multicellular beings. After glucose starvation, yeasts shift from glycolysis to fat metabolism thanks to changes in the histone acetylome [4]. Histone acetyltransferases (HATs) are responsible for transferring an acetyl group from acetyl-CoA to histones and histone acetylation is linked to transcriptional activation. Acetyl-CoA is at the junction of many metabolic pathways, enabling it to bring together gene expression and metabolic state. In the absence of glucose, cells shift to an alternative substrate source that, as we show here, can be FAO. This induces high acetyl-CoA levels [39]. DCA induces sustained activation of PDH, which will also increase acetyl-CoA levels [40]. These two mechanisms lead to an acetyl-CoA excess that could change histone acetylation. Previous studies in vivo suggested that DCA has the potential to induce epigenetic remodeling in the heart, which, at least in part, sustains the molecular basis for its therapeutic effect [40]. The ERK5/MEK5 module plays a central role in endothelial cells [41]. This explains its main role in lipid-related cardiovascular diseases [24,42,43,44], which could be related to ERK5 control of lipoprotein uptake through LDLR [8]. There are other related receptors that bind to native or modified lipoproteins and that influence diverse physiological or pathological processes [34]. Some of them belong to the LRP family, which includes, among others, LRP1, LRP1B, LRP5, LRP6, LRP10, CD36 and CD36L [34]. We show here that all these proteins are regulated by ERK5. Interestingly, CD36 activates ERK5 [23], suggesting the presence of a positive feedback loop that supports continuous lipid intake (Figure 6).

ERK5 not only feeds intracellular lipid availability; it also supports lipid esterification by increasing the expression of ACSL enzymes that allow fatty-acid activation. Finally, ERK5 also regulates expression of the ACADs, which will start FAO. Therefore, ERK5 sustains transcriptome remodeling, allowing long-term fitness in the new environment [1]. Consequently, we observe that relatively long-term DCA incubation leads to a metabolic rearrangement, with cells mainly relying in FAO to obtain energy. In vivo, this would lead to lower circulating lipids and could explain the physiological effects of DCA in patients with combined hyperlipoproteinemia or homozygous familial hypercholesterolemia [6,7,10]. Unfortunately for DCA clinical development, it induces neurotoxicity [12,13].

In summary, we show here that different families of enzymes, which are essential to perform FAO, are regulated by the same kinase pathway, i.e., MEK5/ERK5, that transduces changes from the environment to genetic adaptations.

## 5. Conclusions

Here, we show that glucose starvation or sustained pyruvate dehydrogenase (PDH) activation by DCA induce a large genetic remodeling to propel FAO. ERK5 centralizes remodeling that includes several steps. First, there is an increase in the lipid transport by expression of low-density lipoprotein receptor-related proteins (LRP), e.g., CD36, LRP1 and others. Second, there is an increase in expression of members of the acyl-CoA synthetase long-chain (ACSL) family, which activates FA. Finally, the expression of the enzymes that catalyze the initial step in each cycle of FAO, i.e., the acyl-CoA dehydrogenases (ACADs), is induced. In summary, different families of enzymes, which are essential to perform FAO, are regulated by the same kinase pathway, i.e., MEK5/ERK5, which transduces changes from the environment to genetic adaptations.

## Figures and Tables

**Figure 1 cells-11-01392-f001:**
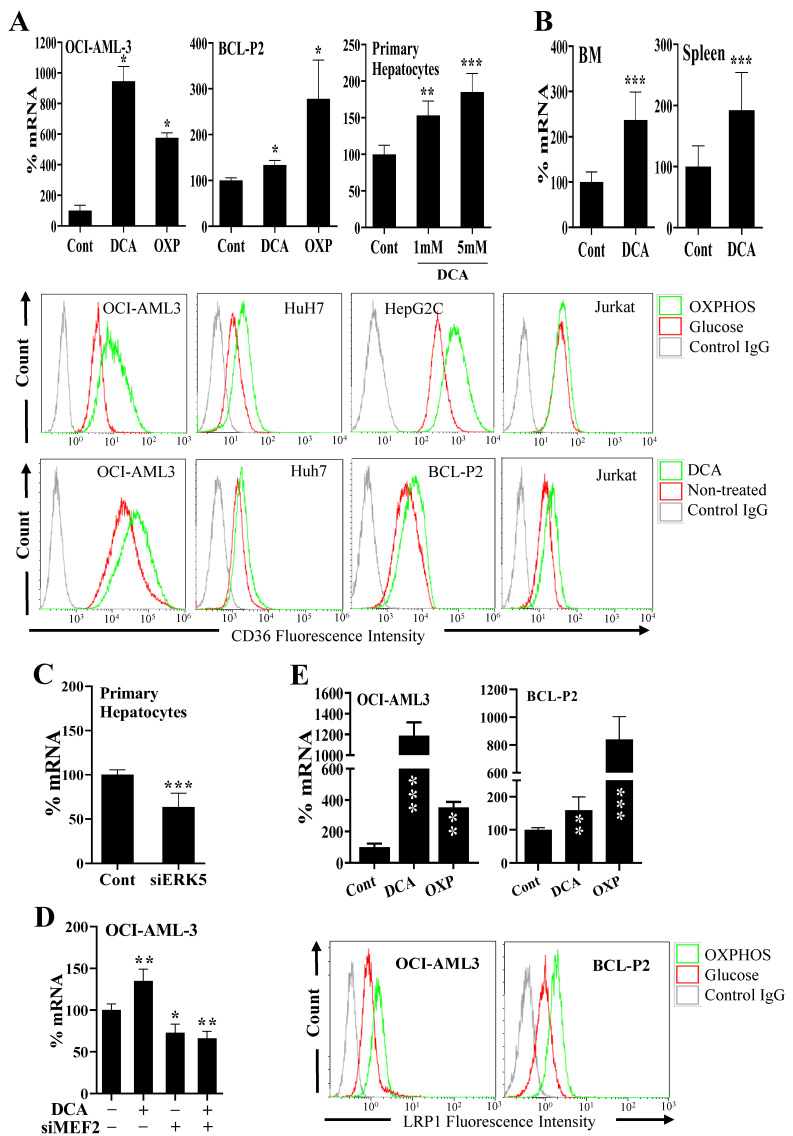
Metabolic changes regulate expression of proteins involved in FA transport through the ERK5/MEF2 pathway. (**A**) CD36 mRNA (up; analyzed by qRT-PCR) or CD36 membrane protein (bottom; analyzed by FACS analysis) were analyzed as described in Materials and Methods section in different cell types treated for 3 days with DCA (5 mM or as indicated) or growing in a glucose-free medium (OXP) that supports OXPHOS for 5–7 days. (**B**) NSG mice were engrafted with primary human AML cells. At day 80 post-graft, they were treated with DCA (n = 4) or left untreated (n = 4). At day 140, mRNA from bone marrow or spleen was isolated and human CD36 mRNA was quantified by qPCR. (**C**,**D**) Cells were treated with siRNA for ERK5 (primary hepatocytes) or MEF2 (OCI-AML cells), and 3 days later we analyzed CD36 mRNA expression. (**E**) Cells were treated as in (**A**) and LRP1 mRNA (up) or LRP1 membrane protein (bottom) were analyzed as described in (**A**). Bar graphs represent means ± SD of at least 3 independent experiments performed in triplicate. * *p* < 0.05, ** *p* < 0.01, *** *p* < 0.005 compared to control cells.

**Figure 2 cells-11-01392-f002:**
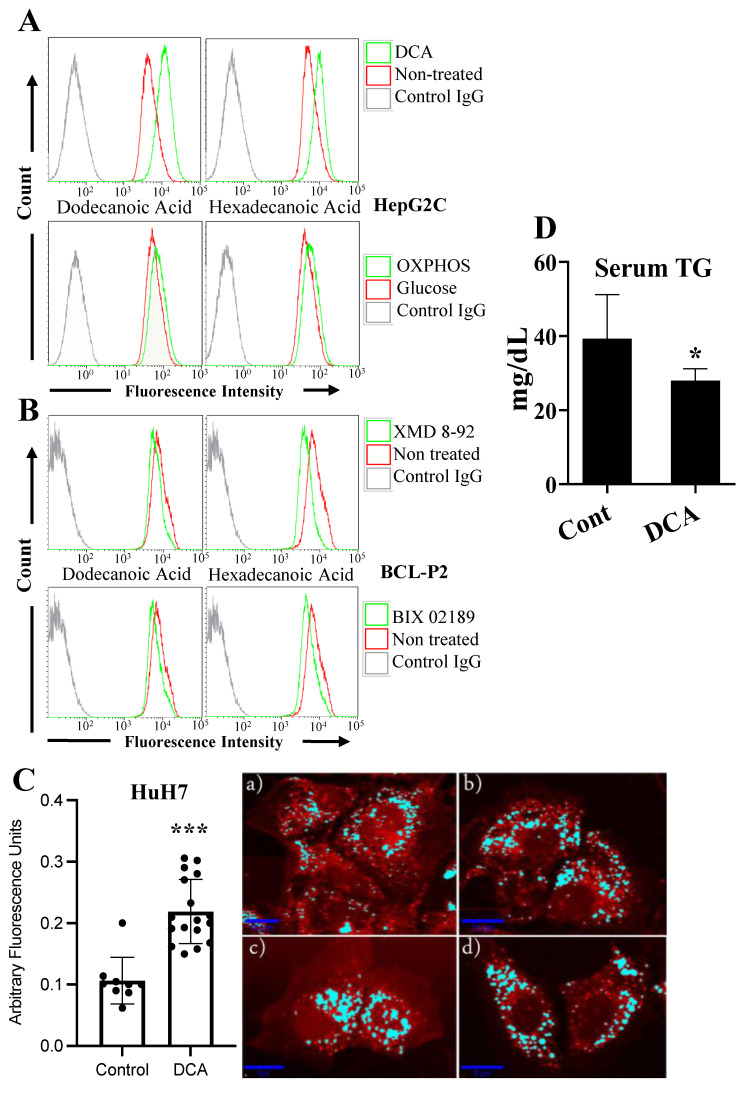
DCA induced intracellular lipid accumulation (**A**) HepG2 hepatocytes were treated with 5 mM DCA for 3 days or grown in a free-glucose medium that supports OXPHOS for 5–7 days. Cells were incubated for 30 min with fluorescent dodecanoic or hexadecenoic FA and the uptake was analyzed by FACS. (**B**) BCL-2 cells were treated with the ERK5 inhibitor XMD 8-92 (10 µM) or with the MEK5 inhibitor BIX 02,189 (5 µM) for 24 h before analyzing FA transport. (**C**) HuH7 hepatocytes were grown for 5–7 days without (**a,b**) or with DCA (**c,d**) as in (**A**) and the intracellular amount of lipids were analyzed by Raman microscopy. (**D**) Mice were treated for 4 weeks with DCA (50 mg/kg per day) and the amount of TG measured by multichannel Hitachi Modular analyzer. Bar graphs represent means ± SD of at least 3 independent experiments performed in triplicate. * *p* < 0.05, *** *p* < 0.005 compare to control cells.

**Figure 3 cells-11-01392-f003:**
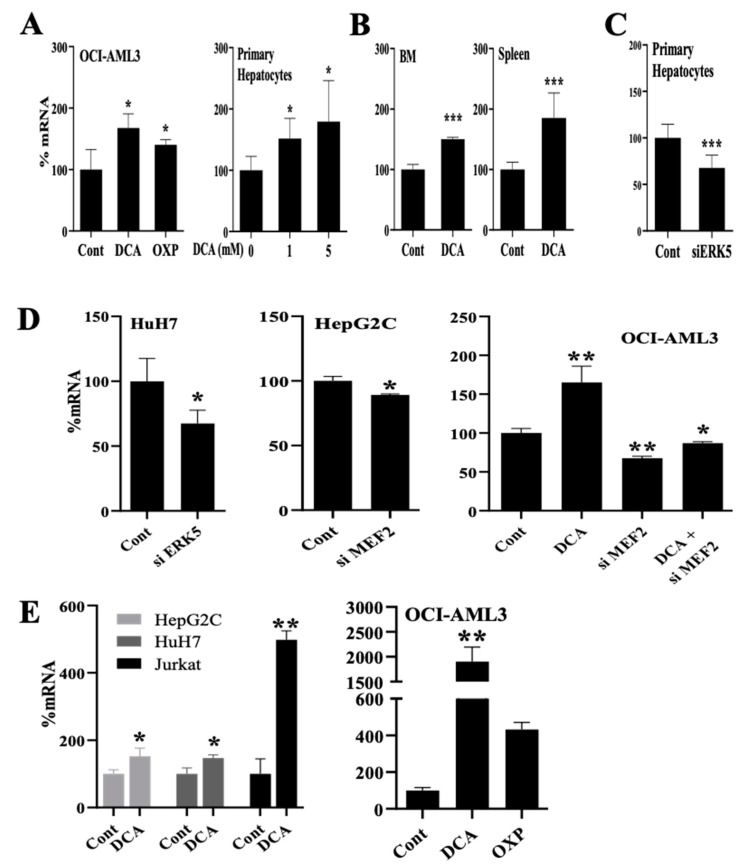
Metabolic changes regulate expression of acyl-CoA synthetase long-chain (ACSL) family members. (**A**) ACSL1 mRNA was analyzed in different cell types treated with DCA (5 mM or as indicated) or growing in a free-glucose medium that induced OXPHOS for 5-7 days. (**B**) NSG mice were grafted with human AML cells and treated with DCA for 16 days before analyzing ACSL1 mRNA expression. (**C**) Primary hepatocytes were treated with siRNA for ERK5, and 3 days later we analyzed ACSL1 mRNA expression. (**D**) Different cell types were treated for 72 h with siRNA for ERK5 or MEF2 and treated with 5 mM DCA for 48 h post-transfection. (**E**) Different cell types were treated as in (**D**) and ACSL6 mRNA expression was analyzed. Bar graphs represent means ± SD of at least 3 independent experiments performed in triplicate. * *p* < 0.05, ** *p* < 0.01, *** *p* < 0.005 compared to control cells.

**Figure 4 cells-11-01392-f004:**
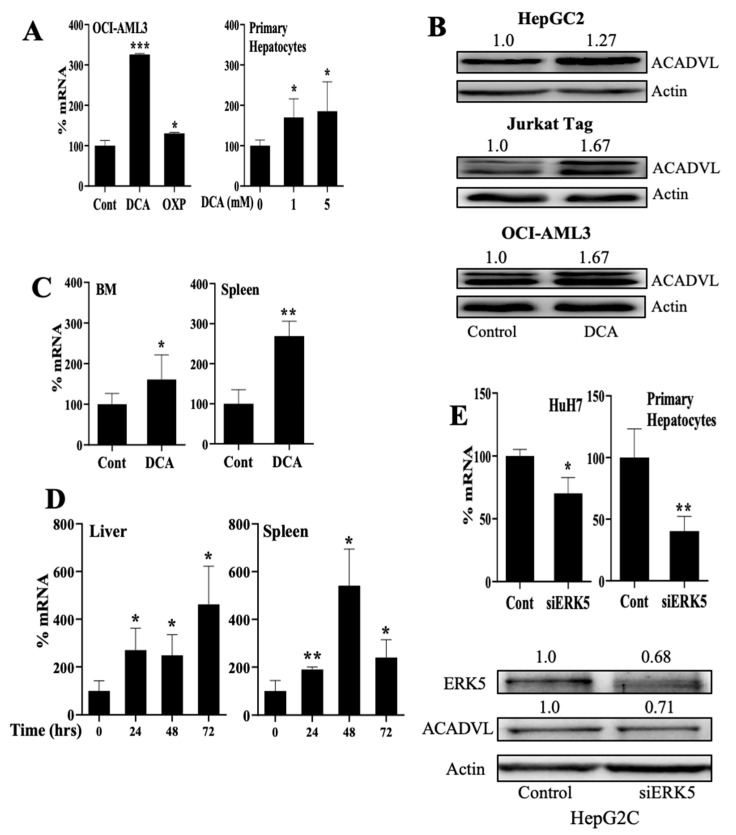
Metabolic changes regulate expression of acyl-CoA dehydrogenases (ACADs) family members. (**A,B**) Different cells were treated with DCA (5 mM or as indicated) or growing in a free-glucose medium that induced OXPHOS for 5–7 days and expression of ACADVL mRNA (**A**) or protein (**B**) were analyzed. (**C**) NSG mice were grafted with human AML cells and treated with DCA for 16 days before analyzing ACADVL mRNA expression. (**D**) acadvl mRNA was analyzed in spleen and liver of mice treated for the indicated times with DCA. (**E**) Different hepatic cells were transfected with siRNA for ERK5 and 3 days later ACADVL mRNA (up) or ACADVL protein (down) were analyzed. Bar graphs represent means ± SD of at least 3 independent experiments performed in triplicate. * *p* < 0.05, ** *p* < 0.01, *** *p* < 0.005 compared to control cells.

**Figure 5 cells-11-01392-f005:**
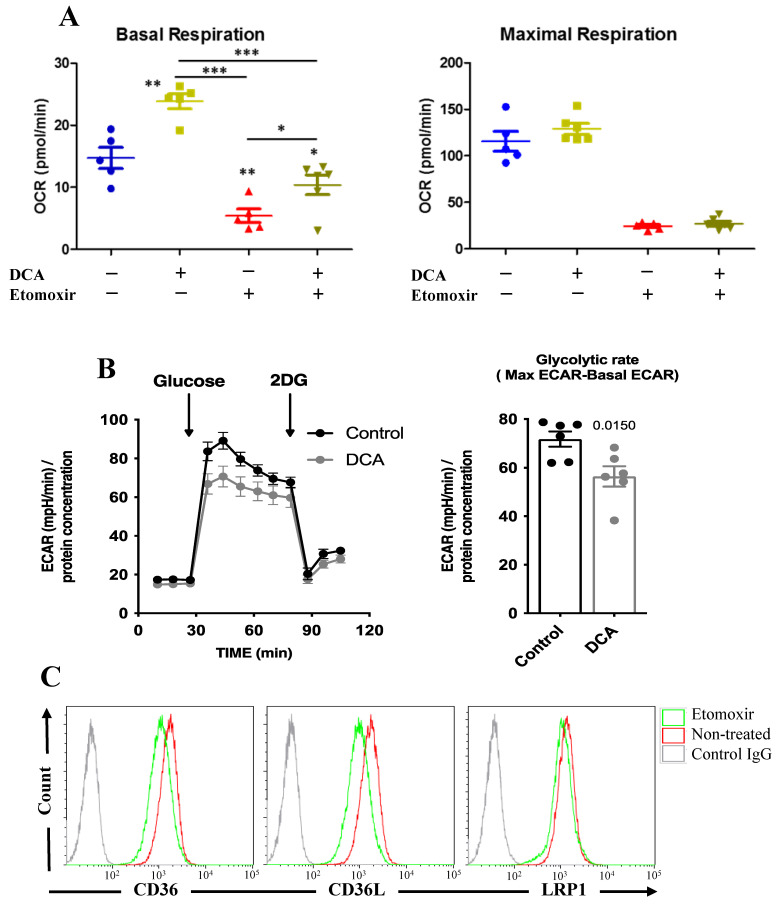
DCA induces FAO. (**A**) HCT 116 cells were incubated with DCA (5 mM) for 3 days and/or etomoxir (5 µM) for 1 h before analyzing the oxygen consumption (OCR) in basal or maximal respiration in a Seahorse analyzer. Bar graphs represent means ± SD of at least 3 independent experiments performed in triplicate. * *p* < 0.05, ** *p* < 0.01, *** *p* < 0.005 compared to non-treated cells. (**B**) HepG2C cells were incubated with 5 mM DCA over 72 h (48 h in a 10 cm plate, and the last 24 h in the seahorse plate). To account for differences in cell proliferation, 60,000 cells were seeded per well in the DCA treatment condition and 50,000 in the control conditions the day before a glycolysis assay in a Seahorse XF24. The day of the assay, the medium was replaced by base DMEM 2 mM Glutamine (no FBS, no Glucose, no Pyruvate). After 3 measurements, 20 mM Glucose was injected. After 6 measurements 100 mM 2DG was injected to completely abrogate glycolysis. ECAR measurements were normalized with protein concentration (measured by BCA in each well). The glycolytic rate was calculated by subtracting maximal ECAR to basal ECAR. The bar graphs represent means ± SD of 6 independent experiments; statistics were performed with Student’s *t*-test compared to non-treated cells. (**C**) Jurkat cells were treated with etomoxir (5 µM) for 24 h before analyzing by FACS the expression of the depicted proteins.

**Figure 6 cells-11-01392-f006:**
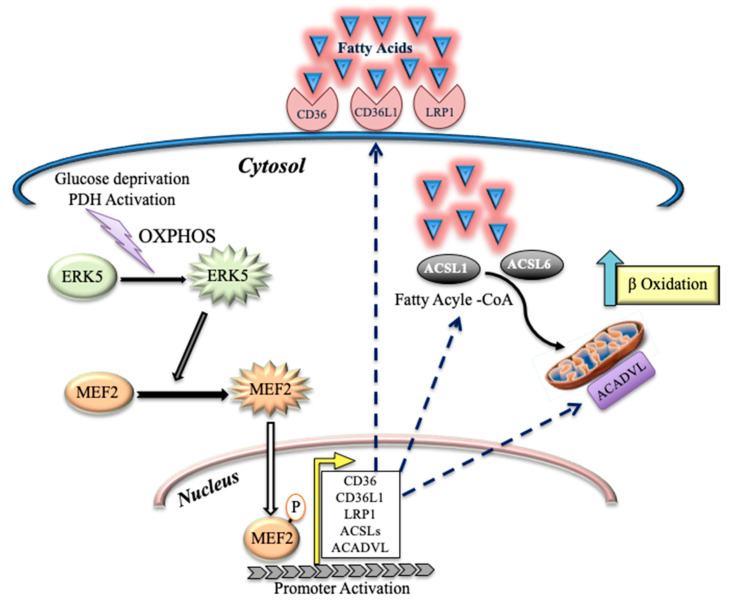
Schematic representation of metabolic remodeling mediated by the activation of ERK5 that is induced by glucose starvation or sustained pyruvate dehydrogenase (PDH) activation.

## Data Availability

Not applicable.

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
