# Peer review of "Glucose Starvation or Pyruvate Dehydrogenase Activation Induce a Broad, ERK5-Mediated, Metabolic Remodeling Leading to Fatty Acid Oxidation"

_cells, 2022, doi:10.3390/cells11091392_

Round 1

Reviewer 1 Report

Language:
Line 48 - inhition to inhibition
Line 60 - "This latter" to MEF2

General notes:
Many figures would benefit from individual data points being shown. For siRNA experiments it is important to know how effective silencing is, and dramatically influences interpretation. If ERK5 is completely silenced and CD36 level only decreases by 30%, that is a much different interpretation than ERK5 is only silenced by 10%. 

Treatment ranges of "several days" is not particularly useful information. Authors should be explicit in methodological details.

A model schematic would also be useful.

Figure 1
The text says CD36 localization to the plasma membrane (Fig 1A). Fig 1A though is qPCR data. The authors should exhibit care when describing the appropriate data in the text.

B: I'm surprised the data is so significant. The DCA treated spleens are about 2 fold increased with an error bar that looks to overlap with the control error bar. It would be useful to see individual data points or a supplement with statistical tests.

C and D: It is important to know how efficient the siRNA transfections were at silencing the target. The authors should include qPCR or western data for ERK5 depletion or MEF2 depletion.

E: The text mentions mRNA but not the LRP1 fluorescence data. Neither does the figure legend.

Figure 5
The authors are commended on nice Seahorse data. However it is surprising to use a cell model which was not previously in the main figures. Why not use HepGC2, OC-AML3, or Jurkat Tag lines? Suspension cells have been used in Seahorse before, and I believe HepG2C should be adherent.

It would be satisfying to see Seahorse data on siERK

Author Response

See reviewer 1 answer in the attached document.

Reviewer 2 Report

In this manuscript, Abrar et al. aim to examine the metabolism regulation mechanisms of DCA treatment or glycolysis inhibition conditions. Using mainly bar graphs and mRNA analysis, the authors claimed that these data explain glycolysis and lipolysis pathways can affect each other, which is already known in the field.

The strength of this work is showing essential details of the PDK1 pathway linked to the membrane regulation pathway in lipid uptake, which offers a bit step that is highly related to the previous publication, Sci Rep 2017,7 10645  doi:10.1038/s41598-017-10339-5. Further demonstration is required while the proof of current findings, including upregulation of CD36 mRNA, ACSL, and ACADs, is still weakly linked to the ERK5 and MEF2 pathway. 

Critiques ï¼š

It is not clear what the primary use for the DCA will be since DCA is a PDK1 inhibitor and stimulate the PDH activity, thus would facilitate glycolysis rather than inhibiting and enhancing glycolysis coupling with mitochondria oxidation. The glycolysis should be checked in detail in addition to the level of lactate (better with metabolic flux analysis).  Other glycolysis inhibitors are suggested, such as 2-DG and UK5099, which would separate glycolysis and mitochondria.

The Raman imaging is impressive and connects to the date of CD36, ACSL et al. While it is not obvious to me great detail is provided in this research article. Will the CD36 or SCARB1 membrane expression level increase? How about their protein trafficking and the relationship of lipid uptake after ERK5 or MEF2 knockdown? Since the OCR is stimulated after DCA treatment,  will lipid droplets' utilization be affected in the cytosol in a long-term time window?

The method is unclear for sodium pyruvate and glutamine composing in the culture medium. The bar graphs that organized in a sloppy way should be improved.

Author Response

See reviewer 2 answer in the attached document.

Reviewer 3 Report

This is an interesting piece of work. I only have minor comments.

  • Figure 1 is not clear to me. The legend does not mention the FACs data and must be more explicit.
  • Line 423: ....does not only feed (not feeds) intracellular lipid availability.

Author Response

See reviewer 3 answer in the attached document.

Round 2

Reviewer 1 Report

I appreciate the changes the authors have made to clarify the text. While I still believe the Seahorse experiment with siERK would be illuminating, I will acquiesce it might be for a future publication.